# Water Sorption by Polyheteroarylenes

**DOI:** 10.3390/polym14112255

**Published:** 2022-05-31

**Authors:** Anatoly E. Chalykh, Tatiana F. Petrova, Igor I. Ponomarev

**Affiliations:** 1Frumkin Institute of Physical Chemistry and Electrochemistry Russian Academy of Sciences (IPCE RAS), 31, Bld.4 Leninsky Prospect, 119071 Moscow, Russia; chalykh@mail.ru; 2A.N. Nesmeyanov Institute of Organoelement Compounds of Russian Academy of Sciences (INEOS RAS), Vavilova St., 28, 119991 Moscow, Russia; gagapon@ineos.ac.ru

**Keywords:** diffusion, sorption of water vapor, diffusion coefficients, polyheteroarylenes, polynaphthoyleneimidobenzimidazole, hydrate numbers of functional groups

## Abstract

The sorption–diffusion characteristics of rigid-chain glassy polymers based on polyheteroarylenes (PHAs) have been studied in a wide interval of relative humidity and temperatures of thermal treatment of the polymer sorbents. Experimental data on water vapor sorption for polynaphthoyleneimidobenzimidazole (PNIB) and its copolymers with different chemical nature have been obtained. Water diffusion coefficients have been calculated, and parameters of their concentration and temperature dependences have been determined. It was found that water molecules sorbed by PNIB and its copolymers are strongly bounded. Water mobile and cluster states depend on the structure of macromolecules and thermal prehistory of polymer sorbents. It is shown that the translational coefficients of water diffusion for all PHAs are in the range from 10^−9^ to 10^−8^ cm^2^/s. The diffusion coefficients also increase slightly with temperature increasing, and their general dependence on temperature is satisfactorily described by the Arrhenius equation. The average activation energy of water diffusion varies from 24.3 to 25.9 kJ/mol. The hydrate numbers of rigid-chain PHAs functional groups have been determined. The above-mentioned results allow us to predict the sorption properties of heterocyclic macromolecular sorbents with complex chain architecture.

## 1. Introduction

Among heat-resistant synthetic polymers, polyheteroarylenes (PHAs) and, in particular, polynaphthoyleneimidobenzimidazole (PNIB) and its block or statistical copolymers occupy a special place [1,2,3].

The interest in this class of high-molecular compounds is not accidental and is caused by their high thermal, fire and radiation resistance, unique physical–mechanical and strength properties [4,5]. PNIB and its derivatives have attracted the attention of polymeric material scientists as potential candidates for creating films and fibers, current source membranes, electroconductive and electro insulating materials, adhesives, nonlinear optical devices, and protective insulating coatings [5,6,7].

The high strength and physicochemical properties of PNIB and its copolymers are traditionally attributed, on the one hand, to the high equilibrium rigidity of the chains (Kuhn segment of PNIB from 25 to 35 nm [5,6]), and on the other hand, to “the strong system of dipole-dipole interactions and hydrogen bonds” formed in the matrix due to the polar carbonyl groups of naphthoyleneimide and –NH– groups of benzimidazole cycles in the polymer units [8,9]. PNIB films are highly resistant to the action of polar organic solvents. Thus, it was found [10,11,12,13] that in such solvents as DMFA, phenol, and chloroform, which are used for orientation stretching of the PNIB films, the physical–mechanical and performance parameters of PNIB films and fibers practically do not change. Nevertheless, for these materials, there is a “sharp” increase in their affinity toward water, which, in our opinion, is quite natural, since the polar functional groups of the naphthoyleneimide and benzimidazole cycles possess high values of hydrate numbers, according to [14].

Unfortunately, there is practically no information on the water sorption of PNIB and its copolymers. At present, only a few works are known [12,13], in which the state of “absorbed” PNIB water was analyzed by FTIR and relaxation spectrometry (DMA) methods. The authors of these studies established that the water molecules sorbed on PNIB are present in three states:-“water molecules forming the first hydrate layer” are firmly bound to the polymer;-“liquid bulk water mode”, filling the “pores” of the polymer and is the least strongly bonded to it;-“structured water mode”, whose molecules are bound together by hydrogen bonds.

However, direct structural–morphological, phase and thermodynamic proofs of the proposed mechanism of water sorption have not yet been obtained.

The aim of this work was to investigate the sorption of water vapor by rigid-chain sorbents such as PNIB and its copolymers with different thermal treatment, the structure and composition of macromolecular chains in order to identify the translational mobility and state of sorbed water molecules in the PHA matrix, to determine hydrate numbers and thermodynamic parameters of interaction in the PNIB–water system.

## 2. Experimental

### 2.1. Objects and Methods

PNIB and its copolymers were synthesized at the A.N. Nesmeyanov Institute of Organoelement Compounds RAS (Moscow, Russia and presented in the form of films that were used as objects of research:-Polynaphthoyleneimidobenzimidazole PNIB
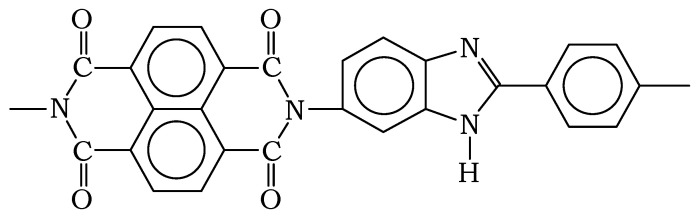
was derived from the dianhydride of 1,4,5,8-naphthalenetetracarboxylic acid (DNTA) and 5(6)-amino-2-(*p*-aminophenyl) benzimidazole (DAB) according to the method [14,15];-Copolynaphthoyleneimidobenzimidazoles of the statistical structure PNI-4:
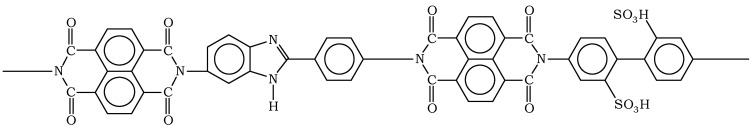
containing 2,2′-disulfobenzidine (DSB) links were prepared by high-temperature polycondensation of DNTA, DAB and DSB. Changing the molar ratio of DAB and DSB regulated the number of links containing “hydrophilic” (acidic) sulfonic groups in the copolymer main chain [9];-Statistically structured PNI-5 copolynaphthoyleneimidobenzimidazoles based on DAB and 3,5-diaminomesitylene (DAM) were prepared similarly to PNIB and PNI-4 [16]:
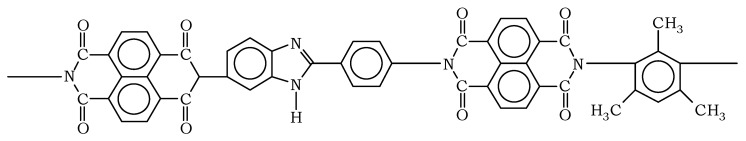


All measurements were performed on isotropic films of PNIB, PNIB−DSB (PNI-4) and PNI−DAM (PNI-5) from 30 to 50 μm thickness obtained from reaction solutions in phenolic solvents (phenol + *p*-chlorophenol) by pouring them onto a glass substrate followed by drying from 10 to 15 h in vacuum at 10^−2^ torr at 413–423 K.

The studies were performed on polymer films sorbent of three types:-PNIB-1, PNIB-4.1 and PNIB-5.1, formed under the above conditions and containing up to 15–18 wt% phenol according to mass term analysis;-Samples of extracted PNIB-1, PNIB-4.1 and PNIB-5.1 in a mixture of acetone–methyl alcohol (PNIB-2, PNIB-4.2 and PNIB-5.2) with a residual phenol content of less than 1 wt%;-Samples of PNIB-2 extracted and vacuum annealed in the free state for 2 h at 543 K PNIB-3.

The molecular weight was determined for all tested samples by the method of flow birefringence [17], which varied in the range from 150 to 250 kDa. The density of PNIB and its copolymers was determined by hydrostatic weighing at 293 ± 1 K in gradient columns filled with a mixture of toluene–carbon tetrachloride. It varied from 1.357 (PNI-4.1 and PNI-5.1) to 1.429 (PNIB-2) and 1.431 (PNIB-3) g/cm^3^, respectively. The glass transition temperature of the films was determined by DMA method in the resonant oscillation mode from 80 to 200 Hz in an argon flow in the temperature range from 77 to 800 K on samples of size 24 × 2 × 0.05 mm. *T_g_* was evaluated by the temperature position of the peak in the temperature dependences of the tangent angle of mechanical losses *tgδ* and the loss modulus *E*’ [18].

### 2.2. Sorption Measurements

The sorption of water vapor by the samples of PNIB and its copolymers was studied by the traditional method on Mc Bain-Bakr vacuum scales [19] with quartz spirals of 1 mg/mm sensitivity and an optical registration system. The sample mass change was determined by stretching a calibrated quartz spiral using a KM-9 cathetometer with an accuracy of 0.01 mm, which provided an accuracy of ±10^−5^ g for the sample mass measurement. All samples were conditioned in a dry desiccator at zero humidity (CaCl_2_) before measurements. All measurements were carried out in the temperature range from 263 to 320 K and relative humidity (*p*/*p_s_*) from 0.10 to 0.98.

The modes of interval sorption were used [19]. At each stage of interval sorption, measurements were performed until the sorption equilibrium was established, which was taken as the state of the sorbent with the sample mass (*M*) unchanged in time at constant *p* and *T*. The value of *M*(*t*) that remained unchanged for twice the time of equilibrium establishment was taken as *M*_∞_. Thus, the obtained data on dependence of sorption capacity of samples *x*/*m* = (*M*_(*t*)_ − *M_o_*)/*M_o_*, g/g on vapor activity were used for the construction of sorption (desorption) isotherms. The work also applied recalculation of the sorbate volume fraction by mass fraction: *C*_2_ = (*φ*/1 − *φ*) (*ρ_water_*/*ρ_polymer_*), where *ρ—*density.

Statistical processing of data on isotherms of sorption and kinetics of establishment of sorption equilibrium, performed in the framework of the traditional approach [19,20], showed that under the chosen conditions of the experiments, the total error in determining sorption capacity is 5%.

To calculate the diffusion coefficients [14,15], we used the traditional Fick’s equations of the kinetics of water vapor sorption by the PNIB film of thickness *L*:(1)MtM∞=γ=4π1/2 (DtL2)1/2
(2)MtM∞ =1−∑{8(2n+1)2π2exp(−D(2n+1)2π2tL2)}
where *M_t_* is the mass of the sample at time *t*; *M_t_* is the time-invariant mass of the sample at constant vapor pressure *p* and temperature *T*. *M*_∞_ was assumed as the value of *M_t_*, which remained unchanged for a time twice as long as the time of equilibrium establishment; *γ* is the degree of filling the sorbent with sorbate, *D* is the partial diffusion coefficient of water vapor, cm^2^/s, and *t* is time. Statistical processing of the experimental data on the sorption kinetics showed that the relative error in determining the diffusion coefficients at the interval method of the experiment was 5%.

### 2.3. Mass-Thermal Analysis

Studies were carried out to determine the temperature intervals of phenol and water desorption remaining after synthesis in PNIB. The work was performed on a Varian MAT 311A (USA) mass spectrometer with an attachment for programmed sample heating and a quartz glass pyrolysis cell connected to a system of direct input to the mass spectrometer ion source. Desorption was carried out in the temperature range from 170 to 473 K in the linear mode of temperature rise at rates of 5 and 20 deg/min. Since it was shown in the preliminary experiments that water desorption from the PNIB occurs rather quickly already at the stage of placing the sample into the lock chamber of the mass spectrometer, special attention was paid to the preliminary saturation of the PNIB with water. For this purpose, the sample, after swelling in water, was quenched by cooling in liquid nitrogen and, in this state, was placed in the pre-cooled mass spectrometer cell. Such a technique made it possible to analyze the gas medium practically at 193 K. The sample water loss did not exceed 2 wt%. The experimental procedure did not differ from the traditional one [9,18] and consisted of registering the intensity of molecular peaks of water and phenol (m/e = 18, m/e = 94, respectively) at a constant rate of cell heating. Since the measurements were performed at different heating rates, the shift of the maximums of water and phenol desorption rates were used to calculate the activation energy of water desorption by Kissinger’s equation [9].

## 3. Results and Discussion

### 3.1. Mass-Thermal Analysis

Figure 1 shows typical thermograms of desorption of low molecular weight substances from pre-water saturated PNIB samples. It can be seen that the kinetic curves show two characteristic maxima along with continuously increasing desorption. One maximum is localized in the region of 368–393 K, the second is localized in the region from 423 to 473 K.

Analysis of the mass composition of the desorption products showed that the first peak is associated with water desorption, while the second is associated with phenol desorption. For extracted PNIB samples, the intensity of the second peak decreases in proportion to the phenol fraction decrease (according to independent chemical analysis). Two circumstances should be noted:-The position and intensity of the phenol desorption process is practically independent of the water saturation conditions of PNIB, which allows us to assume that the thermodynamic compatibility of phenol with this polymer is greater than that of water. Indeed, the activation energy of desorption of phenol is from 75.4 to 83.7 KJ/mol, while that of water is from 46.1 to 50.2 KJ/mol;-Assuming that the desorption of phenol occurs completely at the above temperature range, we estimated the concentration of “tightly bound” water fraction by the ratio of peak areas characteristic of water and phenol, it was found that this value ranged in the range from 1.5 to 2.5 wt%.

### 3.2. Structural Studies

Structural studies were performed using scanning electron microscope JSM-U3 firm “JEOL” (Japan) and transmission electron microscope TEM 301 (Netherlands). The structural etching of the film surface in oxygen high-frequency discharge plasma (oxygen pressure in the etching zone was 4.07 × 10^−5^ kgf/cm^2^ at an accelerating voltage of 1.2 kV and frequency of 10 mHz with electron energy according to the double electric probe from 3 to 5 eV, etching time was 40 min) was used to reveal the supramolecular organization of samples.

Diffractograms of samples were obtained on a diffractometer Dron-3 (IC “Burevestnik”, Moscow, Russia). The images were taken on reflection, using CuK_α_ radiation. The structural and morphological studies showed that the morphology of PNIB and its copolymers is characterized by the presence of anisotropic particles domains with a cross section from 70 to 100 nm. During phenol extraction, the size of the domains increases slightly. This is probably due to some loosening of the matrix. All the examined samples of PHAs were found to be monolithic amorphous structures (Figure 2).

### 3.3. Kinetics of Sorption and Water Diffusion

Figure 3 and Figure 4 show typical interval kinetics of the sorption and desorption equilibrium establishment for PNIB and its copolymers. It can be seen that practically for all PHAs under isobaric–isothermal conditions in the range of vapor activity *p*/*p_s_* from 0 to 0.8, the kinetics of establishment of sorption equilibrium has a Fickian character. The amount of water vapor sorbed or desorbed at the initial parts of the kinetic curves linearly depends on *t*^1/2^; i.e., water vapor diffusion in the early stages takes place as in a semi-infinite medium at the degree of filling of the sorbent *γ* = *M*_(*t*)_/*M_0_* ≤ 0.5. Only at high humidity for annealed samples PNIB-3 was the pseudo-Fickian kinetics observed, which was characterized by the fall of sorption rate at degrees of filling *γ* ≥ 0.7. We assume that under these conditions, structural rearrangements of sorbents are observed as a result of their plasticization by the sorbent.

The sorption curves determined for the polymer sorbents in the same range of water-filling degrees for films of different thicknesses coincide. This fact indicates that the mechanism of water vapor sorption by the studied PHAs is independent of temperature, the presence of residual solvent, and copolymer chain composition.

The shape of the sorption curves is almost insensitive to the concentration dependence of the diffusion coefficient, which is characterized by extreme dependence (Figure 5).

Thus, it can be stated that for the studied glassy rigid-chain polymer sorbents, a distinctive feature is that the water vapor concentration at the outer surface of the sorbate (film) immediately reaches its equilibrium values at any partial vapor pressure and remains constant throughout the sorption or desorption phase.

On the basis of the kinetic sorption curves and using Equations (1) and (2), the concentration dependences of relative partial diffusion coefficients D of water in PNIB and its copolymers were determined (Figure 5). It can be seen that the translational water diffusion coefficients for all PHAs are in the range from 10^−9^ to 10^−8^ cm^2^/s similar to other glassy polymers [21]. With temperature increasing, the diffusion coefficients slightly increase, and their general dependence on temperature is satisfactorily described by the Arrhenius equation. The average activation energy of water diffusion varies from 24.3 to 25.9 kJ/mol. For comparison, the values of diffusion coefficients of nitrogen and argon through the PNIB films were obtained by gas chromatography and the values of activation energy of the noble gases were determined, which were close to the values of activation energy of water molecules (from 16.7 to 20.9 kJ/mol) in the vapor activity region of *p*/*p_s_* ≤ 0.3.

The specific behavior for this group of polymeric sorbents is a decrease in the partial diffusion coefficients in the region of *p/p_s_* ≥ 0.7 ÷ 0.8. In our opinion, it is related to the thermodynamic non-ideality of water solutions in PNIB, whose intensity increases when the systems approach the limiting concentration. We use the sorption isotherm which relates the vapor activity *p*/*p_s_* to the solution composition as the concentration dependence of the partial diffusion coefficient *D = D** (d*lnp*/*p_s_*/d*lnφ*). The self-diffusion coefficients of sorbed water vapor *D**, which characterizes the translational mobility in the PNIB matrix and its copolymers, were calculated [21]. The dependence *lgD** − *φ*_1_ (Figure 5) is concave with respect to the concentration axis. The curvature becomes more noticeable as the degree of sorbent filling increases, which is associated with the plasticizing effect of the polymer sorbate. Obviously, this effect can be ascribed to a non-significant increase in the mobility of the PHAs segments. On the basis of relaxation spectrometry data, it was shown earlier [5,6] that sorbed water affects low- and high-temperature transitions in PNIB, shifting them to the region of low temperatures. For these polymers, a sharp decrease in the elastic modulus and a decrease in the glass transition temperature with wetting were detected.

Thus, the results of the sorption studies confirm the relaxation spectrometry data. When increasing the amount of sorbed water (*p*/*p_s_* > 0.8), a plasticizing effect of PHAs is clearly observed.

### 3.4. The Sorption of Water Vapor by PNIB. Influence of Thermal Background

Using the experimental data, the isotherms of water vapor sorption by PNIB-1, PNIB-2 and PNIB-3 samples are calculated and show a pronounced *S*-shaped form (Rogers type III sorption isotherms). The fact of anomalously high sorption capacity of PNIB samples in relation to water vapor attracts special attention. Thus, x/m is ≥ 15 wt% at *p*/*p_s_* = 0.9 and 293 K. This value reaches the level of cellulose and significantly exceeds the values of sorption capacity typical of aliphatic and aromatic polyamides [22,23]. According to this feature, PNIB-1 and PNIB-2 can be attributed to the class of hydrophilic polymers, which does not correspond to PNIB-3.

Another distinctive characteristic of this class of sorbents is a significant dependence of the sorption capacity on their thermal background (Figure 6). Thus, the extraction of residual solvent (phenol) from PNIB-1 resulted, as a rule, in an increase in its sorption capacity by water vapor by 1.5 times. It should be noted that the phenol content in PNIB-2 does not exceed 1.5 wt%. In contrast, the thermal annealing of films at 775 K (near *T_g_*) have a sorption capacity of PNIB samples at all values of vapor activity that is two times less than that of PNIB-1 treated under normal conditions.

The introduction of 2,2′-disulfobenzidine and 3,5-diaminomesitylene fragments into the PNIB chain does not principally change the sorption character.

As in the case of PNIB, the isotherms of water sorption by copolymers are *S*-shaped (Roger’s type III sorption isotherms [24]); for the extracted samples of copolymers, their sorption capacity increases, on average, by 1.7, and thermal annealing leads to a decrease in the sorption isotherms. The sorption–desorption hysteresis for these sorbents is also weakly expressed.

Despite the fact that the copolymers are in the glassy state throughout the *p/p_s_* interval, there is a dependence of the sorption capacity on the structure of the PHAs monomer link for them. Comparing the isotherms in Figure 7, it can be seen that the introduction of –SO_3_H groups into the polymer chain leads to an increase in the sorption capacity by 12 wt% over the whole vapor activity interval, whereas the introduction of 3,5-diaminomesitylene into the chain slightly reduces the sorption capacity as compared to the initial sample.

The change in their sorption capacity with temperature is also unusual for PNIB (Figure 8). For PNIB-1 and PNIB-2, the sorption capacity of these polymers decreases with increasing temperature, while for PNIB-3, it remains practically unchanged. This trend of change in solubility with temperature is characteristic of systems with a lower critical mixing temperature (LCMT), a lower critical solution temperature of the components (LCST), and according to Papkov’s suggestion, it can be regarded as a “lateral critical temperature” [23].

For PNI-4 copolymers containing sulfonic groups in their composition, the sorption capacity increases with increasing temperature, in contrast to PNIB (Figure 7). For PNI-5.1 copolymers containing methyl-groups in their macromolecules, the sorption capacity is either constant or the temperature coefficient of water solubility in a relatively narrow temperature range changes its sign to the opposite.

As noted above, the sorption isotherms are reproducible in contrast to the desorption isotherms, which was found in repeated cycles of sorption–desorption. As shown in Figure 9, a weak but well reproducible sorption–desorption hysteresis is observed for these sorbents. We suggest that the presence of hysteresis loops can be explained by the non-equilibrium supramolecular structure of polymers and the relaxation processes in them. At the stage of sorption during plasticization, a more equilibrium structure is traditionally formed in the sorbent with lower internal stresses, which reappear during desorption. The reversibility of sorption–desorption isotherms testifies to the achievement of equilibrium states by the system.

The experimental data were interpreted within the framework of a double sorption model [21], considering the total amount of absorbed water as the sum of water fraction sorbed on the available polymer active centers and water fraction dissolved and migrating in the polymer matrix. Figure 10 shows schematically the possible localization of sorbed water molecules on the functional groups of the chain and in the inter-chain space [5,6]. For comparison, the possible localization of phenol molecules is presented.

From the formal point of view, the *S*-shaped isotherms can be represented as a superposition of the Langmuir isotherm (Equation (3)) and the Flory–Huggins isotherm (Equation (4)) characterizing, respectively, fractions of sorbed water molecules localized on active centers and modes of dissolved water molecules freely migrating through the volume of sorbate:(3)$$\phi_{1L}=\phi_{1L}^{\infty }\frac{k\left(p/p_{S}\right)}{k\left(p/p_{S}\right)+1},$$
(4)$$p/p_{S}=\exp\left[\ln\left(\phi_{1F}\right)+\left(1+\frac{1}{r}\right)\phi_{2F}+\chi \;{\left(\phi \right)}_{2F}^{2}\right]$$
where $\;\phi_{1L}^{\infty }$ is the maximum Langmuir sorption, *k* is the sorption equilibrium constant, and *p*/*p_s_* is the vapor activity. Hereinafter, subscript 1 refers to the solvent and subscript 2 refers to the polymer. The constants $\;\phi_{1F}$ and *k* were calculated from the initial section of isotherms. The constant *χ* was calculated from the final part of the isotherm by the “gradient descent” method. The calculated values of the constants $\;\phi_{1F}\;\phi_{1L}^{\infty }$, *k* and *χ* are shown in Table 1.

Examples of the decomposition of sorption isotherms are shown in Figure 11. It can be seen that for all investigated PNIB under different temperature–humidity conditions, there is good agreement between the experimental values of sorption isotherms and the total sorption isotherms calculated by the Langmuir and Flory–Huggins component contributions. The Langmuir component most fully describes the change in the sorption capacity of PNIB from *p*/*p_s_* 0 to *p*/*p_s_* from 0.4 to 0.5 in the original sample and to *p*/*p_s_* from 0.3 to 0.4 in the extracted one. With temperature increasing, it shifts toward higher moisture content. At *p*/*p_s_* ≥ from 0.4 to 0.5, the Langmuir component reaches its constant value, which in absolute value is between 1.5 and 3.5 wt% (40 ÷ 60% of the total amount of sorbed moisture). The Langmuir component for PNIB depends significantly on the thermal prehistory of the sample.

The extraction of phenol leads to an increase in sorption capacity while annealing to its decrease. The calculated interaction heats of water with the polymer active centers in this section are 10–11 kJ/mol, which is significantly lower than the interaction parameters of “strongly bound” water according to the mass-term analysis. One can assume that water molecules sorbed by the mechanism of localized adsorption on the functional groups of the polymer are present in two states that differ from each other in the strength of binding to the active centers. Probably, this indicates the different availability of PNIB active groups to the interaction with water, which, in turn, is determined by the peculiarities of its structural and morphological organization. The PNIB link size is 1.8 nm according to X-ray diffraction analysis.

The analysis of sorption isotherms reveals the mechanism of interaction of water molecules with the active functional groups of PGAs (according to Langmuir isotherm). The Flory–Huggins isotherms, which describe the dissolution of water in the polymer matrix, show themselves most fully at *p*/*p_s_* = 0.5. Like the Langmuir component, these fragments of the sorption isotherms depend on the thermal prehistory of the samples, decreasing as the sorbent density increases during thermal annealing. Note that for the annealed sample, the sorption capacity according to Flory–Huggins is comparable with the sorption on the available PNIB active centers, and the character of the sorption isotherms is closer to the Henry isotherm; that is, we can assume that for PNIB-3, the character of the sorption process is described by the Langmuir + Henry combination. A comparison of PNIB-3 isotherms with PNIB-1 and PNIB-2 (Figure 11) shows that such a transition from Langmuir + Flory–Huggins to Langmuir + Henry is associated only with changes in the total amount of sorbed moisture, i.e., a property of the Flory–Huggins isotherm.

In contrast to Langmuir, the sorption capacity according to Flory–Huggins decreases with increasing temperature; the interaction parameters of Flory–Huggins components also decrease. This course of the temperature dependence testifies *χ* to the existence of the LCST of PNIB in water. Estimates of the critical temperature values corresponding to *χ* − *T_k_* made by extrapolation of the *T_k_* dependence showed that the LCST of the system lies in the interval of *T* from193 to 198 K.

The values of the heat of sorption calculated from the Flory–Huggins isotherm range from 6 to 8 kJ/mol, which is somewhat less than their values in the Langmuir section, i.e., these thermodynamic parameters are somewhat lower than the values for irreversibly bound water.

Additional information on the state of water molecules in aramid sorbents was obtained using the Starkweather equations [19]. It is known that a quantitative characteristic of the formation of associates–clusters by sorbed water molecules of associates–clusters in the polymer matrix is the clustering integral
(5)$$\frac{G_{11}}{v_{1}}=-\left(1-\varphi_{1}\right)\left[\frac{\partial \frac{a_{1}}{\varphi_{1}}}{\partial a_{1}}\right]-1,$$
where *v*_1_, *φ*_1_ and *a*_1_–are the molar volume, volume fraction and activity of the component, respectively. The intensity of clustering in solutions is determined by how much the ratio *G*_11_/*v*_1_ exceeds the value −1. Recall that for an ideal solution, *G*_11_/*v*_1_ = −1. For the systems we studied, the boundary value of relative humidity is *p*/*p_s_*, which characterizes the beginning of clustering, *p*/*p_s_* ≌ 0.6 (marked with an arrow in Figure 12).

The average number of water molecules in the clusters *N_c_* was determined by the equation
(6)$$N_{c}=\varphi_{1\;}\left(\frac{G_{11}}{v_{1}}+1\right)+1$$

It was shown that the average number of molecules in the cluster changes from 2 to 3 with a change in the value of relative humidity *p*/*p_s_* from 0.5 to 0.9, which is in good agreement with the infrared spectroscopy data obtained by the author [9]. Probably, the clustering process is accompanied by a change in the intermolecular distance, the appearance of a plasticizing effect, and violation of the liquid crystal order, which is accompanied by an increase in water vapor sorption and structural amorphization. Indeed, for all sorbents, a weakly pronounced sorption–desorption hysteresis is observed, indicating the restructuring of their structure in the process of sorption. Using the clasterization integral and sorption isotherms corresponding to Flory–Huggins, we determined the range *p*/*p_s_* in which clusters of sorbed water molecules are formed. These clusters are formed at a moisture from 0.5 to 0.6 and represent associations of water molecules.

The obtained isotherms were also analyzed in the framework of the hydrate number conception. The methodology for determining hydrate numbers did not differ from that proposed by Van Krevelen [11]. The specificity of our approach is that only a certain section of Flory–Huggins sorption isotherms were used in the calculations. The results of the calculations are presented in Table 2.

Combination of these data with the previously published Van Krevelen data makes it possible to predict the sorption properties of macromolecular sorbents with complex chain architecture.

## 4. Conclusions

The water vapor sorption isotherms of polynaphthoyleneimidobenzimidazole (PNIB) and its copolymers, depending on their chemical structure, composition, temperature and thermal treatment have been studied. It was established that for all polymer sorbent sorption isotherms, an *S*-shaped character has been strongly pronounced. Within the frame of the double-sorption model, it was shown that water molecules sorbed in a matrix of rigid-chain PNIB and its copolymers are present in three states: firmly bonded state localized on functional groups in cavities of free volume; mobile state dissolved in the sorbent volume; and clustered state distributed in the overall polymer volume. Hydrate numbers of polymer functional groups were determined, the pair parameter of Flory–Huggins interaction was calculated, and its temperature dependence was determined. Diffusion coefficients of sorbed water molecules and diffusion activation energy were obtained. All results are in good agreement with FTIR and DMA relaxation spectrometry data.

## Figures and Tables

**Figure 1 polymers-14-02255-f001:**
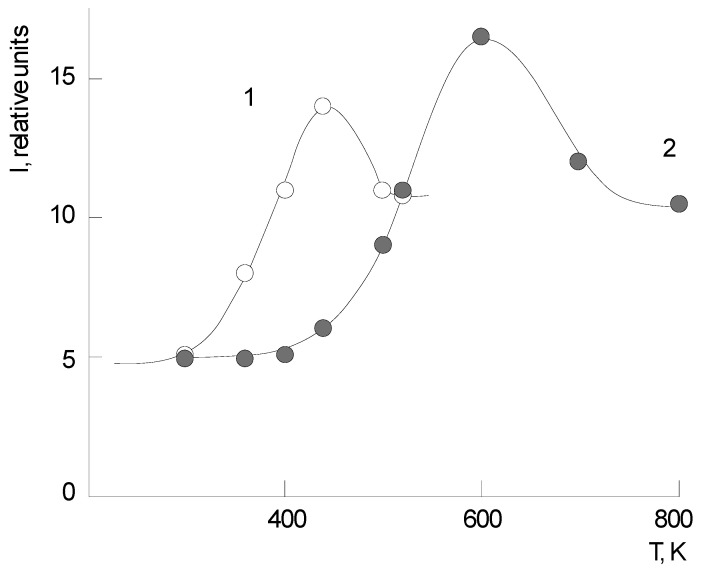
Desorption thermograms for water (1) and phenol (2) according to mass- thermal analysis.

**Figure 2 polymers-14-02255-f002:**
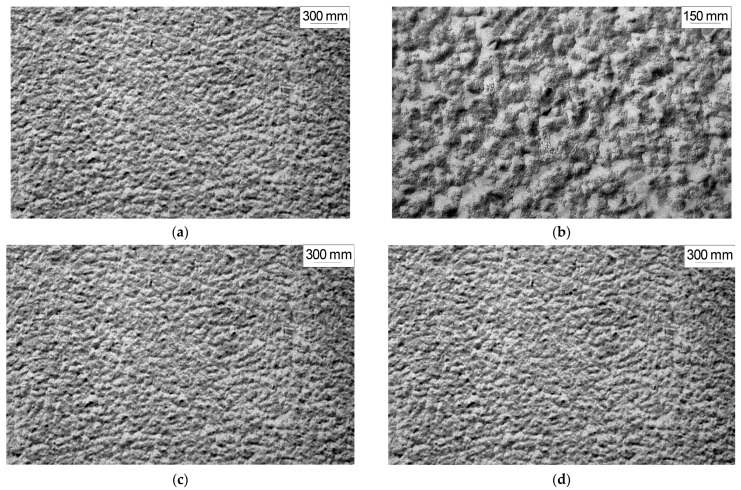
Domain structure of polymers films: (**a**) PNIB-1, (**b**) PNIB-2, (**c**) PNIB-4.1, and (**d**) PNIB-5.1.

**Figure 3 polymers-14-02255-f003:**
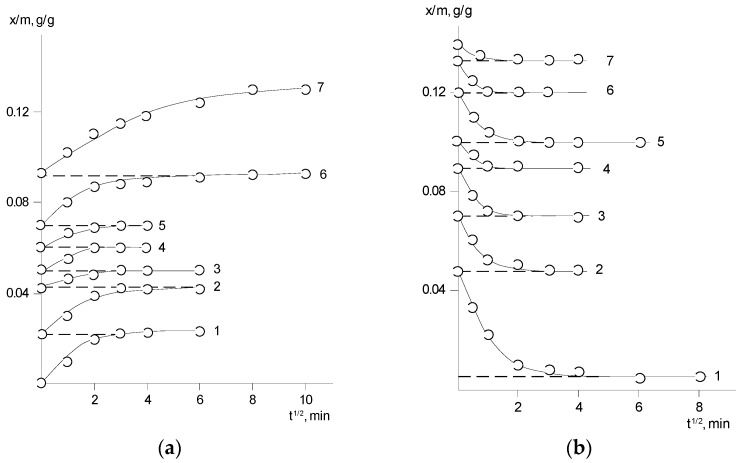
Kinetic curves of water sorption (**a**) and desorption (**b**) in PNIB-1 at 30 °C in the *p/p_s_* intervals for a (sorption): (1) 0–0.06; (2) 0.06–0.17; (3) 0.17–0.27; (4) 0.27–0.42; (5) 0.42–0.5; (6) 0.5–0.72; (7) 0.72–0.87; for b (desorption): (1) 0.12–0; (2) 0.24–0.12; (3) 0.36–0.24; (4) 0.50–0.36; (5) 0.66–0.50; (6) 0.82–0.66; (7) 0.87–0.82.

**Figure 4 polymers-14-02255-f004:**
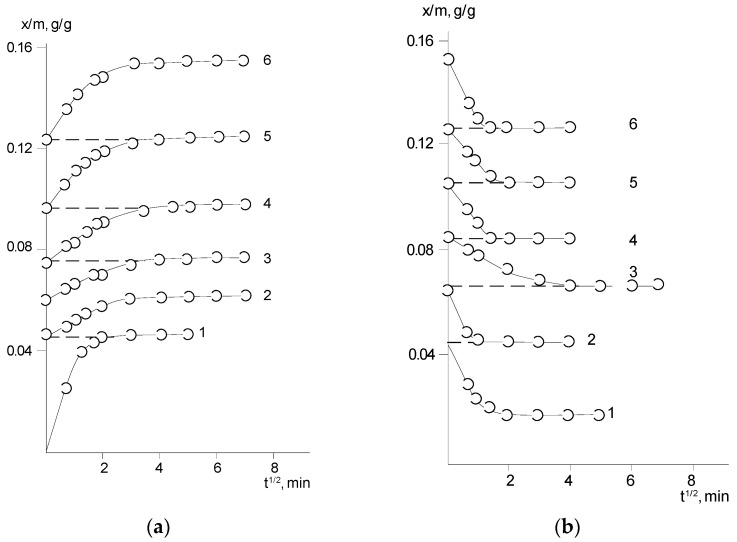
Kinetic curves of sorption (**a**) and desorption (**b**) of water in PNI-4.1 at 313 K. in the *p*/*p_s_* intervals for (**a**) (sorption): (1) 0–0.17; (2) 0.17–0.29; (3) 0.29–0.42; (4) 0.42–0.55; (5) 0.55–0.69; (6) 0.69–0.81; for (**b**) (desorption): (1) 0.11–0; (2) 0.21–0.11; (3) 0.35–0.21; (4) 0.48–0.35; (5) 0.65–0,48; (6) 0.81–0.65.

**Figure 5 polymers-14-02255-f005:**
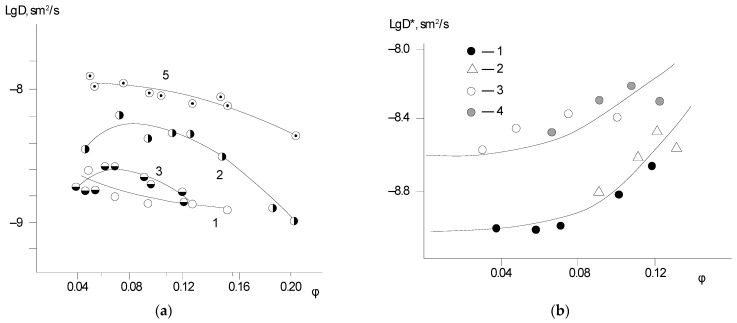
Concentration dependence of the partial diffusion coefficient *D* (**a**) and self-diffusion *D** (**b**) of water in PNI-4.1 (1), PNI-4.2 (2), PNI-5.1 (3), PNIB-1 (4), PNI-5.2 (5) at 303 K.

**Figure 6 polymers-14-02255-f006:**
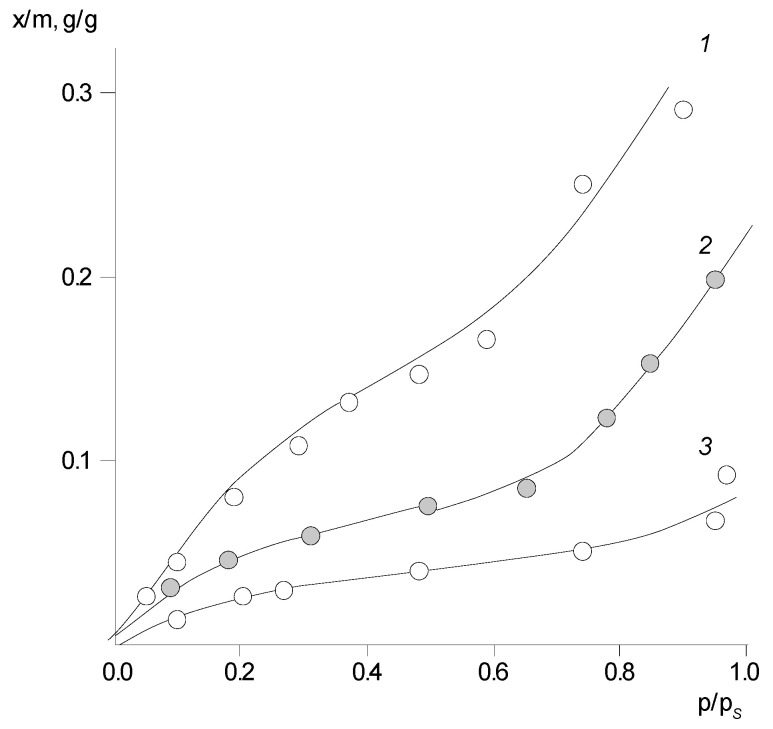
Isotherms of water vapor sorption by PNIB of different thermal treatment at 293 K: 1—PNIB-2, 2—PNIB-1, 3—PNIB-3.

**Figure 7 polymers-14-02255-f007:**
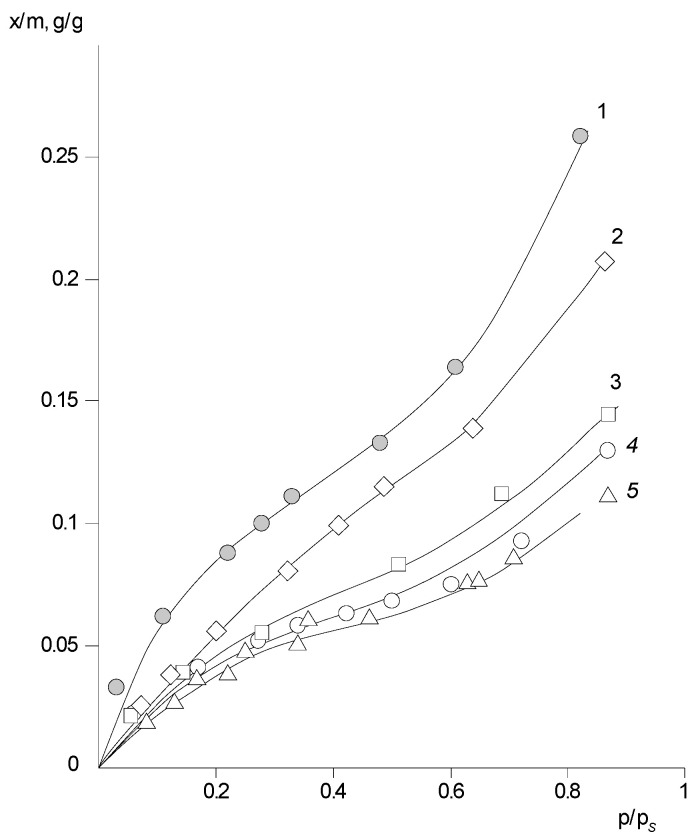
Isotherms of water vapor sorption of PNIB and PNIB copolymers at 303 K: 1—PNIB-4.2; 2—PNIB-5.2; 3—PNIB-4.1; 4—PNIB-1; 5—PNIB-5.1.

**Figure 8 polymers-14-02255-f008:**
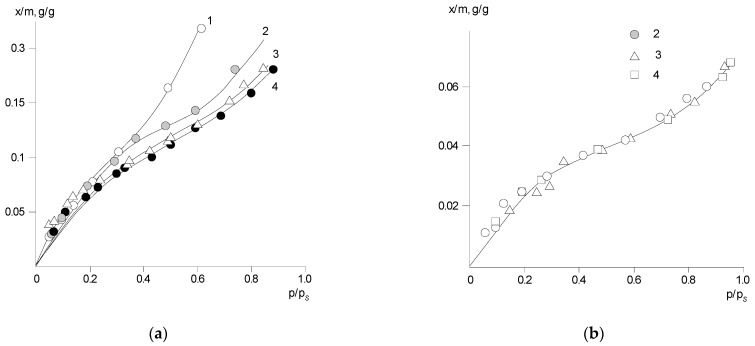
Isotherms of water vapor sorption by PNIB-2 (**a**) and PNIB-3 (**b**) at different temperatures: 1—283 K, 2—293 K, 3—303 K, 4—213 K.

**Figure 9 polymers-14-02255-f009:**
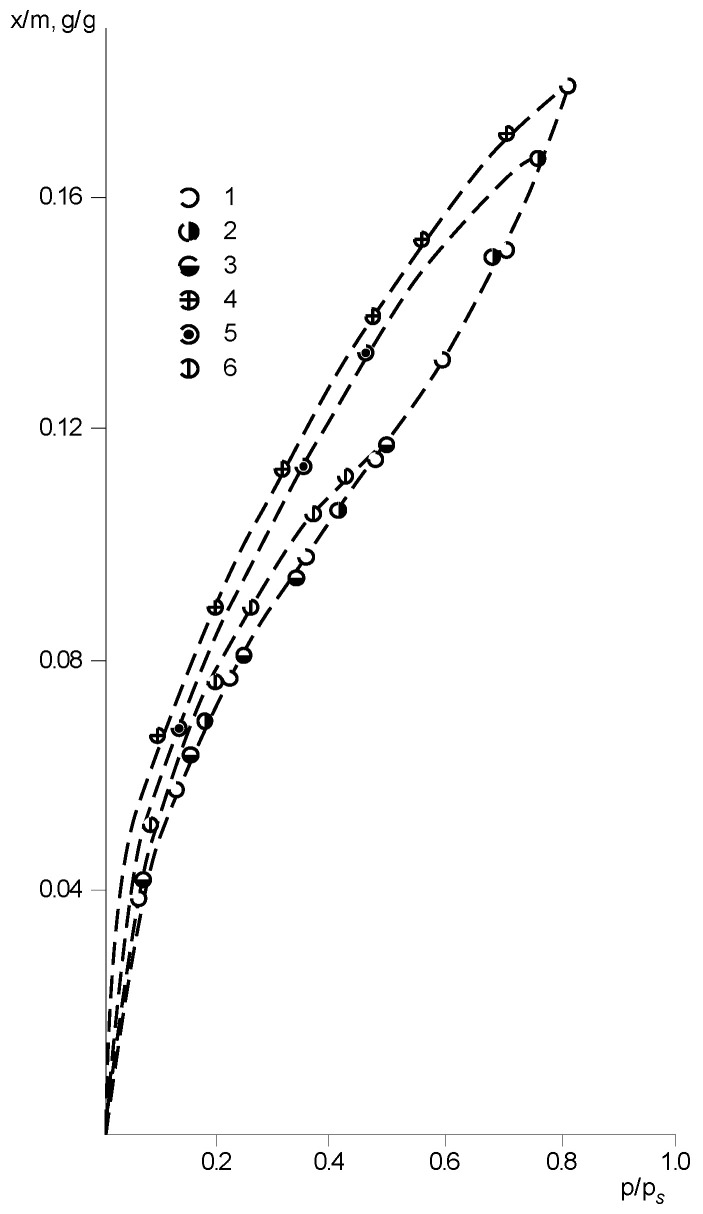
Sorption–desorption hysteresis of water vapor in PNIB-2 at 303 K. Sorption (1–3), desorption (4–6).

**Figure 10 polymers-14-02255-f010:**
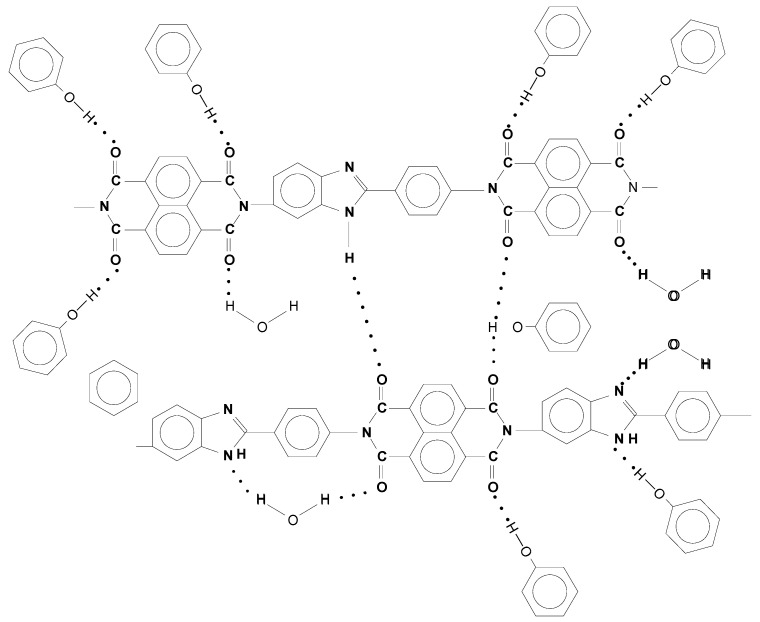
The solvation scheme of PNIB-2 with phenol and water.

**Figure 11 polymers-14-02255-f011:**
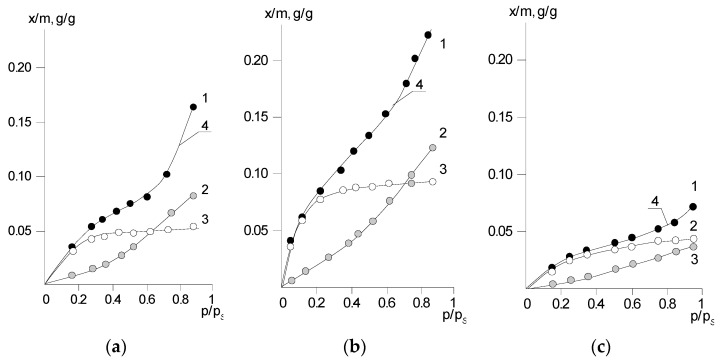
Decomposition of experimental isotherms of water vapor sorption PNIB-1 (**a**), PNIB-2 (**b**), PNIB-3 (**c**) at 303 K into components: 1— experimental isotherm ; 2—Flory–Huggins isotherm; 3—Langmuir isotherm; 4—sum of the components.

**Figure 12 polymers-14-02255-f012:**
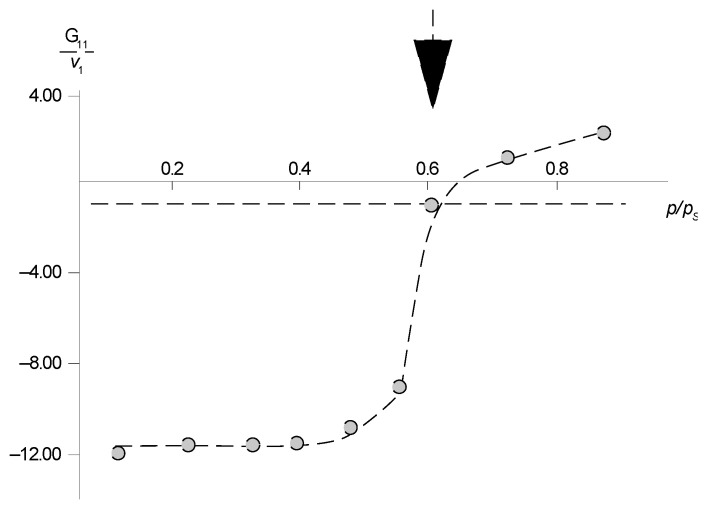
Integral of water clustering in PNIB-1 at 303 K. Explanations in the text.

**Table 1 polymers-14-02255-t001:** The calculated values of the constants $\phi_{1F}\;\phi_{1L}^{\infty }$, *k* and *χ*.

Polymer	Temperature, K	$\phi_{1L}^{\infty }$	*k*	χ
PNIB-1	283	0.0230	166.4	0.80
	303	0.0475	10.58	1.62
	313	0.0514	7.68	1.63
PNIB-2	283	0.0604	8.49	0.52
	293	0.1084	5.52	1.24
	303	0.0875	10.6	1.32
	313	0.0918	6.17	1.25
PNIB-3	297	0.0420	4.27	2.37
	313	0.0291	7.88	2.19
PNI-4.1	298	0.0525	14.4	1.84
	303	0.0560	7.05	1.53
PNI-4.2	303	0.0971	16.01	1.65
PNI-5.1	298	0.0437	3.73	1.72
	303	0.0468	7.09	1.72
	313	0.0399	4.98	1.65
PNI-5.2	303	0.0790	4.41	1.21

**Table 2 polymers-14-02255-t002:** Hydrate numbers of functional groups.

Functional Groups	*p*/*p_s_*			
	0.3	0.5	0.7	0.9
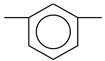	0.585	1.040	1.667	2.527
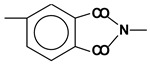	0.039	0.089	0.125	0.180
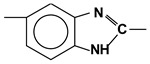	0.054	0.120	0.225	0.320
–SO_3_H–	0.395	0.720	1.188	1.780
–SO_2_–	0.225	0.427	0.676	1.134
–CONH–	0.113	0.224	0.339	0.512
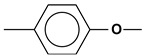	-	0.0015	-	0.0061
–OH	0.182	0.353	0.532	0.818

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
