# Peer review of "Water Sorption by Polyheteroarylenes"

_polymers, 2022, doi:10.3390/polym14112255_

Round 1

Reviewer 1 Report

The manuscript is missing line numbers so referring to areas within the text that require correction is a hurdle.

The manuscript requires extensive English language editing as well as appropriate formatting. These suggest that care was not taken in reviewing the manuscript before submission. 

There are many inconsistencies in the manuscript. The authors should go through and correct all such instances. For example, Figure is written in full in some parts but abbreviated in some portions of the manuscript (page 12 and 15). Also page 5: "thermodesorption" and "thermode-sorption".

What do the authors mean by "prodiges" in the abstract?

Why do the authors insert the figure for the compounds mid-text? Instead, they should appear before or after the paragraph with an appropriate caption included like all figures or schemes in scientific journals. The authors are strongly advised to refer to the instructions for authors.

Figure formatting is sub-standard. Improve the quality of all figures and be sure to align and embed any inclusions to the figures. 

Figure 4a has some entries missing. Correct it.

Equation 3 and 4: Why are the equation numbers duplicated? Remove all instances of equation numbers in-situ the equation and maintain the ex-situ ascribed ones. 

Follow the format for generating tables for this journal.

Page 3: i) What is Cl4. If the "4" is suppose to be a subscript, do so and correct all other such instances in the manuscript;  ii) "24*2*0.05 mm". Use the multiplication sign; iii) "5  8%."; this is unscientific. Is that an error range the authors are attempting to indicate? 

Rewrite the entire conclusion.

Author Response

Thank you for your review of our manuscript. We have answered each of your points below.

1) The manuscript is missing line numbers so referring to areas within the text that require correction is a hurdle.

Response: We would like to apologise for any inconvenience in reviewing the article. Polymers has no strict requirements for submitting an article in the LaTex system or other formats. When submitting in another format, the editorial board usually converts the article to the desired format on its own.

2) The manuscript requires extensive English language editing as well as appropriate formatting. These suggest that care was not taken in reviewing the manuscript before submission. 

Response: The manuscript text has been edited and formatted.

3) There are many inconsistencies in the manuscript. The authors should go through and correct all such instances. For example, Figure is written in full in some parts but abbreviated in some portions of the manuscript (page 12 and 15). Also page 5: "thermodesorption" and "thermode-sorption".

Response: The inconsistencies on  page 12 and 15 have been corrected. The term "thermodesorption" has been replaced by "desorption".

4) What do the authors mean by "prodiges" in the abstract?

Response: Corrected in the annotation to «сopolymers».

5) Why do the authors insert the figure for the compounds mid-text? Instead, they should appear before or after the paragraph with an appropriate caption included like all figures or schemes in scientific journals. The authors are strongly advised to refer to the instructions for authors.

Response: The position of the figures in the text of the article has been adapted according to the instructions for authors.

6) Figure formatting is sub-standard. Improve the quality of all figures and be sure to align and embed any inclusions to the figures. 

Response:In accordance with the comments of the reviewer, figures are presented at a high degree of resolution

7) Figure 4a has some entries missing. Correct it.

Response: Figures 4a and 4b are corrected according to the figure legend.

8) Equation 3 and 4: Why are the equation numbers duplicated? Remove all instances of equation numbers in-situ the equation and maintain the ex-situ ascribed ones. 

Response: The equations are numbered and corrections have been made to the text of the article.

9) Follow the format for generating tables for this journal.

Response: The format of the tables has been adapted to the instructions for the authors.

10) Page 3: i) What is Cl4. If the "4" is suppose to be a subscript, do so and correct all other such instances in the manuscript;  ii) "24*2*0.05 mm". Use the multiplication sign; iii) "5 ¸ 8%."; this is unscientific. Is that an error range the authors are attempting to indicate? 

Response: The Cl4 formula has been deleted from the text of the article, similar cases in the manuscript have been corrected.

11) Rewrite the entire conclusion.

Response: The text of the conclusion has been supplemented and corrected:

            The water vapor sorption isotherms of polynaphthoyleneimidobenzimidazole(PNIB) and its copolymers, depending on their chemical structure, composition, temperature and thermal treatment have been studied. It was established that for all polymer sorbent sorption isotherms S -shaped characterhasbeen strongly pronounced.. Within the frame of double-sorption model it was shown that watermolecules sorbedin a matrix of rigid-chain PNIB and its copolymers are present in three states: firmly bonded state localised on functional groups in cavities of free volume; mobile state dissolved in the sorbent volume and clustered state distributed in the overall polymer volume. Hydrate numbers of polymer functional groups were determined, the pair parameter of Flory-Huggins interaction was calculated and its temperature dependence was determined.Diffusion coefficients of sorbed water molecules and diffusionactivation energy were obtained. All results are in good agreement with FTIR and DMA relaxation spectrometry data.

Reviewer 2 Report

The authors reported a study dealing with the water vapour sorption properties of rigid-chain glassy polymers based on polyheteroarylenes differing in terms of relative humidities, temperatures of measurement and thermal prehistory of sorbents. Water diffusion coefficient were calculated as well as the number of hydrates included in copolymers during sorption process. The topic is interesting, but the way of presenting results and interpretation lust be reviewed.

In my opinion, the paper could be eligible for publication in Polymers after major modifications. Below, some specific comments are included.

  1. Could you explain the term ‘thermal prehistory’ and its concept with respect to your samples.
  2. Could you recall the structures of the tested copolymers, because it’s crucial to clearly understand the purpose and interpretation of your work.
  3. Equation 1 referring to Langmuir-type sorption is lacking.
  4. What do you mean by “weakly expressed extreme dependence” ?
  5. The text needs to be reviewed carefully because unclear.

Abbreviations need to be explained:

HN, LCMT, LCTR,

Minor points:

Floeri-Huggins, Flory-Haggins

Thermode-sorption

masstermograms

Author Response

Thank you for your review of our manuscript. We have answered each of your points below.

1) The authors reported a study dealing with the water vapor sorption properties of rigid-chain glassy polymers based on polyheteroarylenes differing in terms of relative humidities, temperatures of measurement and thermal prehistory of sorbents. Water diffusion coefficient were calculated as well as the number of hydrates included in copolymers during sorption process. The topic is interesting, but the way of presenting results and interpretation lust be reviewed.

Response: The text of the article has been amended and corrected.

2) In my opinion, the paper could be eligible for publication in Polymers after major modifications. Below, some specific comments are included.

Response: The text of the article has been amended and corrected.

3) Could you explain the term ‘thermal prehistory’ and its concept with respect to your samples. Response: The term "thermal prehistory " is replaced by "thermal treatment".

4) Could you recall the structures of the tested copolymers, because it’s crucial to clearly understand the purpose and interpretation of your work.

Response: Please pay attention, that section 3.2 shows the results of structural and morphological studies of the domain structure of copolymers, Figure 2 shows microphotographs of the samples.

5) Equation 1 referring to Langmuir-type sorption is lacking.

Response: The numbers of the equations have been corrected: "Langmuir isotherm (equation 3) and Flory-Huggins isotherm (equation (4)"

6) What do you mean by “weakly expressed extreme dependence” ?

Response: The text has been amended to read: " extreme dependence. "

7) The text needs to be reviewed carefully because unclear.

 Response: We have tried to take all the comments into account and made changes and clarifications to the text

8) Abbreviations need to be explained:

HN, LCMT, LCTR,

Response: Changes have been introduced: HN - deleted, "LCMT" and "LCTR" replaced by "LCST"

9) Minor points:

Floeri-Huggins, Flory-Haggins

Response: Corrected: Flory-Huggins

10) Thermode-sorption

Response: The term has been replaced by "desorption"

Response: The term has been replaced by "termograms

Round 2

Reviewer 2 Report

Although the topic is interesting, the manuscript need to be carrefully proofreading. Some sentences are written without verbs. Some spaces are lacking. In addition, interpretation of some points need to be clarified to improve the quality of the paper.  

Author Response

We are delighted to thank you for revisiting and reviewing our article. In accordance with your requests, the text has been thoroughly verified by all co-authors and improvements and corrections have been made. Unfortunately, we can see that when working with different versions of Word, spaces disappear in some places and symbols in formulas are changed. We hope that after correcting the editor and saving the text in PDF format, these problems will be resolved.